# Factors Influencing Level and Persistence of Anti SARS-CoV-2 IgG after BNT162b2 Vaccine: Evidence from a Large Cohort of Healthcare Workers

**DOI:** 10.3390/vaccines10030474

**Published:** 2022-03-18

**Authors:** Cristina Costa, Enrica Migliore, Claudia Galassi, Gitana Scozzari, Giovannino Ciccone, Maurizio Coggiola, Enrico Pira, Antonio Scarmozzino, Giovanni La Valle, Paola Cassoni, Rossana Cavallo

**Affiliations:** 1Microbiology and Virology Unit, University Hospital Città Della Salute e Della Scienza di Torino, 10126 Turin, Italy; rossana.cavallo@unito.it; 2Clinical Epidemiology Unit, University Hospital Città Della Salute e Della Scienza di Torino and CPO Piemonte, 10126 Turin, Italy; emigliore@cittadellasalute.to.it (E.M.); cgalassi@cittadellasalute.to.it (C.G.); giovannino.ciccone@unito.it (G.C.); 3Hospital Medical Direction, Ospedale Molinette, University Hospital Città Della Salute e Della Scienza di Torino, 10126 Turin, Italy; gscozzari@cittadellasalute.to.it (G.S.); ascarmozzino@cittadellasalute.to.it (A.S.); glavalle@cittadellasalute.to.it (G.L.V.); 4Occupational Medicine Unit, University Hospital Città Della Salute e Della Scienza di Torino, 10126 Turin, Italy; mcoggiola@cittadellasalute.to.it (M.C.); enrico.pira@unito.it (E.P.); 5Pathology Unit, Department of Medical Sciences, University of Turin, 10126 Turin, Italy; paola.cassoni@unito.it

**Keywords:** SARS-CoV-2, serology, mRNA vaccination, health care workers

## Abstract

We aimed at evaluating quantitative IgG response to BNT162b2 COVID-19 vaccine among health care workers (HCW), and exploring the role of demographic, clinical, and occupational factors as predictors of IgG levels. On May 2021, among 6687 HCW at the largest tertiary care University-Hospital of Northwestern Italy, at a median of 15 weeks (Interquartile range-IQR 13.6–16.0) after second-dose, serological response was present in 99.8%. Seropositivity was >97% in all the subgroups, except those self-reporting immunodeficiency (94.9%). Overall, the median serological IgG value was 990 BAU/mL (IQR 551–1870), with most of subjects with previous SARS-CoV-2 infection or with shorter time lapse (2–8 weeks) between vaccination and serology with values in the highest quintile (>2080). At multivariable analysis, significant predictors of lower values were increasing age, male, current smoking, immunodeficiency, recent occupational contacts, and increasing time lapse from vaccination; conversely, previous infection and recent household contacts were significantly associated with higher IgG levels. Subjects with previous infection kept a very high level (around 2000 BAU/mL) up to 120 days. These results, besides supporting a high serological response up to 4–5 months, suggest predictive factors of faster decay of IgG levels that could be useful in tailoring vaccination strategies.

## 1. Introduction

The relevant impact of Severe Acute Respiratory Syndrome coronavirus 2 (SARS-CoV-2) on global health determined an unprecedented effort for rapid developing and delivering of vaccines. The COVID-19 mRNA vaccines were the first to receive emergency use authorization from the Food and Drug Administration (FDA) and the European Medicines Agency (EMA). In particular, in Italy, the vaccination campaign started at the end of 2020 with the administration of BNT162b2 vaccine (Pfizer/BioNTech) to healthcare personnel and elderly individuals in nursing homes. In 2020, we conducted an observational study on SARS-CoV-2 IgG seroprevalence in relation to clinical, demographic, and occupational factors among health care, administrative, and service personnel, at a large tertiary care university hospital in Turin [1]. Currently, the dynamics and magnitude of serological response in vaccinated individuals are being monitored in terms of levels and duration of anti-spike IgG antibodies. Herein, we report data on qualitative and quantitative SARS-CoV-2 antibody response in this cohort, at a median of 15 weeks after vaccination, to explore factors potentially affecting magnitude and duration of immune response. 

## 2. Materials and Methods

### 2.1. Study Design and Population

The study was conducted at the University Hospital Città della Salute e della Scienza di Torino (CSS), located in Turin, the largest tertiary care Public Utility in North Western Italy.

Since 27 December 2020, COVID-19 vaccine (BNT162b2 mRNA vaccine) has been offered to all workers of CSS. Vaccinations (both doses) took place mostly (88%) in January–February 2021. During the month of May 2021, all CSS workers were invited to attend a serological survey (post-vaccination survey) aimed at evaluating the dynamics and magnitude of IgG response after vaccination. Recruitment was on a voluntary basis, and all the subjects were asked to sign an informed consent and complete a questionnaire before blood collection. The questionnaire investigated demographic data, medical history, occupational data, contacts at risk, and evidence of previous SARS-CoV-2 infection. The study was approved by CSS Institutional Review Board (protocol n. 0046457 27th April 2021). 

### 2.2. Serological Assay

Serological data on serum specimens were studied by the LIAISON^®^ SARS-CoV-2 Trimeric IgG assay chemiluminescent immunoassay (CLIA) (Diasorin, Saluggia, Italy), following the manufacturer’s instruction and using the LIAISON^®^ XL Analyzer [2]. The principal components of the assay are magnetic particles (solid phase) coated with recombinant trimeric SARS-CoV-2 spike protein [3] and mouse monoclonal antibodies to human IgG linked to an isoluminol derivative (conjugate). The assay evaluates the presence of antibodies to SARS-CoV-2, including neutralizing antibodies. Antibody concentrations were calculated by the analyzer and expressed as Binding Antibody Units (BAU/mL), and calibrated with the First WHO International Standard for anti-SARS-CoV-2 immunoglobulin 20/136 [4], allowing for a qualitative grading of the results: <33.8 BAU/mL considered as negative; ≥33.8 BAU/mL as positive. The upper limit for quantification of antibody levels, with no further sample dilution, was 2080 BAU/mL, while the lower detection limit was 4.81 BAU/mL.

### 2.3. Definition of SARS-CoV-2 Infections

Exposure to SARS-CoV-2 before serological test (from March 2020 to May 2021) was defined by the occurrence of any of the following: (a) positivity to molecular assay in upper respiratory specimens (nasopharyngeal swab); (b) any previous positive serological assay, as previously reported [1]; (c) a history of Hospital admission due to COVID disease.

### 2.4. Statistical Analysis

Baseline characteristics of participants and positivity of serological test are summarized as absolute and relative (percentage) frequencies, or as means and standard deviations. Quantitative variables with non-normal distribution (including IgG levels) are reported as medians and interquartile ranges (IQR). To evaluate potential individual and occupational predictors of IgG levels after vaccination, we performed a multivariable ordinal logistic regression model using the quintiles of IgG distribution, considering the highest category as reference. The estimated Odds ratios (OR), with 95% confidence intervals (95%CI), express the risk of belonging to a lower quintile of the IgG distribution, after accounting for all other variables. Distance between vaccination and serological test was calculated as the number of days (or weeks) between the date of the second vaccine dose (first dose for those who received only one dose) and the date of the serological test. Statistical analyses were performed by Stata 15.1 software (StataCorp LP, College Station, TX, USA).

## 3. Results

The cohort study flow is reported in the appendix (Appendix A). Briefly, a serological test was available for 7726 workers (69.5% of the 11115 CSS workers). The analysed population included 6687 subjects that received (by May 2021) at least one dose of vaccine, filled in the questionnaire, and performed the serological test at least 14 days after vaccination. The included subjects had a median age of 49.5 years (IQR 38.7–56.5), 75% were females, 35% were nurses, 25% physicians, and 11% health care assistants (Table 1); most (79.4%) attended the previous serological survey performed on May–June 2020, after the first pandemic wave in Northern Italy [1]. The large majority (6556, 98%) received both doses of vaccine. 

At a median distance of about 15 weeks (IQR 13.6–16.0 weeks) between vaccination and serological evaluation (Table 2), a serological response was present in almost all subjects (99.8%); only 12 subjects out of 6687 (0.18%) had an IgG value below the positivity cut-off of 33.8 BAU/mL, and only 4 of them had an IgG value below the detection limit of 4.81 BAU/mL.

Prevalence of sero-positivity was higher than 97% in almost all the subgroups examined (Table 1, Table 2 and Table 3), with the only exception of subjects that self-reported an immunodeficiency (94.9%).

In the overall population, the median serological value was 990 BAU/mL (IQR 551-1870) (Table 1), with a certain degree of variability among subgroups. As expected, median serological values greater than 2080 BAU/mL (i.e., in the highest quintile of the overall distribution, see Appendix A) were observed in subjects who experienced a previous SARS-CoV-2 infection and in those with a shorter time lapse (from 2 to 8 weeks) between vaccination and serological test (Table 2). Lower serological values were observed among current smokers (Table 1), subjects that self-reported an immunodeficiency or a kidney disease (Table 3), and for those with a longer time lapse (more than 16 weeks) between vaccination and serological test (Table 2). No substantial differences were observed in both positivity and IgG values among AB0 blood groups (Table 1). 

We investigated several demographic, clinical, and occupational factors to explore whether they were associated with lower serological values post-vaccination, using multivariable analysis in order to adjust for eventually unequal distribution of factors among different subgroups. The adjusted associations between these factors and lower serological values (measured as odds ratios of belonging to a lower quintile of the distribution), are reported in Figure 1 and in Appendix A. Significant predictors of lower serological values are increasing age, male gender, being a current smoker, having an immunodeficiency, and having had recent (in 2021) contacts with infected colleagues in the workplace; furthermore, lower serological values increased by an OR of 1.30 (95%CI 1.27–1.33) for each week of time lapse from vaccination. Conversely, a previous SARS-CoV-2 infection and recent (in 2021) household contacts significantly reduced the risk of belonging to lower quintiles, i.e., are associated with higher IgG levels. 

To better evaluate IgG decline after vaccination, we analysed serological values over time separately for subjects without (n.5454) and with (n.1221) a SARS-CoV-2 infection at any time before serological test. Subjects without previous SARS-CoV-2 infection show a slow, but progressive decline in IgG levels starting from 60 days after vaccination, while subjects with previous SARS-CoV-2 infection tend to keep a very high IgG level (around 2000 BAU/mL) up to 120 days after vaccination (Figure 2). Similarly, we observed a more pronounced reduction over time in IgG levels among those reporting an immunodeficiency compared to those not reporting this disorder (Appendix A). 

Our study design does not allow to accurately estimate the incidence of breakthrough SARS-CoV-2 infections in this cohort, since only a fraction of HCW working in at risk wards undergo periodical monitoring with PCR. However, we investigated by means of a questionnaire the occurrence of a positive swab and of COVID symptoms up to May 2021. Out of 6687 vaccinated HCW, only 49 (0.7%, equivalent to an estimated incidence of 0.66 per 10,000 person/day at risk) had a positive swab more than 14 days after the complete vaccination (2nd dose), and 43% of them did not report any symptoms.

## 4. Discussion

In this study, we describe the results of serological evaluation performed at approximately 4 months post-vaccination on a cohort of almost 6700 hospital workers that can be considered representative of anti-spike IgG response to BNT162b2 vaccine in an adult healthy population. The first relevant finding of this study is that >99% of subjects developed and maintained a good serological response to vaccination at least up to 4 months. This rate of seropositivity is similar to those reported in recent publications, such as the study by Eyre and coll. Ref. [5] with 98.9% of >3600 HCW in United Kingdom, the study of Lustig and coll. Ref. [6] with 98.4% of >4000 HCW in Israel, and the study by Kontou and coll. with 99.8% of vaccine antibody response [7]. In other studies on general population, IgG antibody generation following Pfizer/BioNTech vaccine was similarly high, ranging from 94.2% to 98.36% [8,9]. Data on efficacy of BNT162b2 mRNA COVID-19 vaccine evidenced a 95% efficacy in preventing COVID-19 [10]; this efficacy was confirmed in the updated analysis up to 6 months after the administration of the second dose [11]; our preliminary results on breakthrough infections up to 4–5 months post-vaccination appear in line with these studies. 

Considering the extremely low number of subjects without seroconversion observed in our study, no inference can be made about possible relation between demographic, clinical, and occupation features and rate of seronegativity. Hence, we evaluated determinants potentially predicting quantitative antibody responses in our study, which is one of the largest cohorts of health care workers.

In agreement with previous reports, our study supports a role for age in the development and amount of antibody response. Moreover, this significant relation between lower serological values and increasing age is observed also at different time lapses from vaccination, including the first weeks [5,6,12,13,14,15,16], up to 3–6 months such as in our and other studies [17,18]. 

Another demographic feature that we found associated with lower serological levels is male gender (adjusted for all other variables), a finding that deserves further investigations. Several studies evidenced a lower antibody response to BNT162b2 vaccine in men than in women [6,7,12,13,14,15,16,17,19]. Some studies suggested that this difference could be influenced by the serological assay used for analysis. Indeed, this difference in antibody level according to sex was observed in several studies using various commercially available serological assays, including the test used in the present study.

We observed significantly lower serological values among current smokers in comparison to never smokers (OR 2.00, 95% CI 1.79–2.25), in line with other studies that investigated this factor [17,20]. Interestingly, this finding is consistent with the low seroprevalence among smokers observed in the first phase of our study on HCW, after the first pandemic wave [1], and suggests an impairment by tobacco smoke on humoral response [21] both to infection and to vaccination. 

Concerning the impact of other personal features, some studies evidenced a significant association between obesity and a general low seroconversion upon some vaccine administration [22], an increased risk of infection even in the presence of strong serological response [23], and an inverse association between lower antibody levels following SARS-CoV-2 vaccine and waist circumference [20]. Nevertheless, in our study, no significant association was found between serological response and Body Mass Index (BMI). This result is similar to that found by others [13,20]. Although these discrepant results among studies could be due to different assays used as well as the timing of serology, Watanabe and Coll. Ref. [20] underline that BMI poorly reflects actual body fat excess, whereas waist circumference better describes obesity. This issue should be more appropriately investigated as it has already been reported that a higher ratio of visceral fat in relation to total adipose tissue was superior to BMI also in predicting COVID-19 morbidity and mortality [24].

Among potential predictors of antibody response, we investigated the role of several diseases or health conditions, including autoimmune diseases, immunodeficiencies, chronic diseases, and neoplasms; however, we failed to find any significant association with the exception of immunodeficiency. Patients with common variable immunodeficiencies and other primary antibody defects evidence an impaired antibody response to infection and immunization [25,26]. In our study, patients with self-reported immunodeficiency evidenced an increased risk of lower serological values of more than 2-fold, and seropositivity was found in 94.9% of 78 subjects in contrast with 97–100% of individuals with other diseases/disorders. In the series by Pulvirenti and coll. Ref. [26], SARS-CoV-2 infection primed a more efficient classical memory B cell response, whereas the BNT162b2 vaccine induced non-canonical B cell responses in patients with immunodeficiency. In addition, natural immunity following infection was boosted by subsequent immunization, suggesting the possibility to further stimulate the immune response by additional vaccine doses in immunodeficiency disorders.

Considering the impact of previous infection with SARS-CoV-2 on serological response, as previously reported [5,19,27,28,29,30,31,32], we confirmed that this was associated with significantly higher antibody levels, with approximately half of the subjects displaying values exceeding the upper limit of detection and a remaining 20% in the immediately lower quintile. This finding is relevant, considering that our study was performed at 4–5 months post-vaccination, whereas most studies evaluated individuals at 3 to 7 weeks post-vaccination. For example, in the study by Ferrari and coll., in almost 99% of the cases, antibody titers exceeding the upper limit of detection were detected at 21 days after the first dose [32]. The overall available evidence strongly supports that a significant difference, both in magnitude and in persistence, exists in post-vaccination response between subjects with and without previous SARS-CoV-2 infection. Seemingly, a single dose in previously infected individuals results in antibody responses close to those seen after a two-dose vaccination administered to SARS-CoV-2-naïve subjects. 

Potential exposure to the virus, documented by reported household contacts, was associated with higher IgG levels in this study. A major role of household contacts in comparison to other potential contacts was already observed for serological data pre-vaccination [1]. Conversely, in this phase we did not observe higher IgG levels for HCW reporting contacts with positive patients or colleagues, suggesting the efficacy of standardized preventive measures adopted in the hospital setting after the very first phase of the pandemic. 

As the humoral responses to SARS-CoV-2 infection have been proved to confer a relevant protection against morbidity and mortality from subsequent infections [33,34,35], evaluation of antibody responses after vaccination can provide relevant information in terms of long-term effectiveness [36]. Based on our results, seropositivity persists up to 4–5 months post-vaccination in >99% of health care workers; however, a significant decrease in antibody levels can be observed as the time lapse between vaccine administration and serological test increases. Further evaluation at different time points from vaccination could allow to better define the pattern of this decline and reduction of effectiveness. However, available data have been considered sufficient to support the strategy of administering a third booster dose, as to date performed in Italy and in other countries. 

Our study has some limitations, including the lack of measurement of neutralizing antibody levels. However, the serological assay used in this study has been shown to closely correlate with the titer of neutralizing antibodies [2]. A potential limit is that clinical information were collected through a questionnaire, with self-reported diseases and disorders, with the only exception of previous SARS-CoV-2 infection, that was ascertain through a hospital database collecting results of molecular swabs or previous serological positivity.

## 5. Conclusions

In conclusion, we presented data from one of the largest vaccinated health care worker cohort to date, initially surveyed for seropositivity and then tested for serological response and related factors following administration of BNT162b2 COVID-19 vaccine. Overall, our results confirm the high serological response to this vaccine, with duration and persistence of response up to 4 months. These results confirm the preventive strategy established by Italy and other countries. The identification of predictive factors of antibody response and duration could support more tailored vaccine strategies, although further studies on cellular immunity are needed to better clarify the host role in the modification of vaccine strategy effectiveness.

## Figures and Tables

**Figure 1 vaccines-10-00474-f001:**
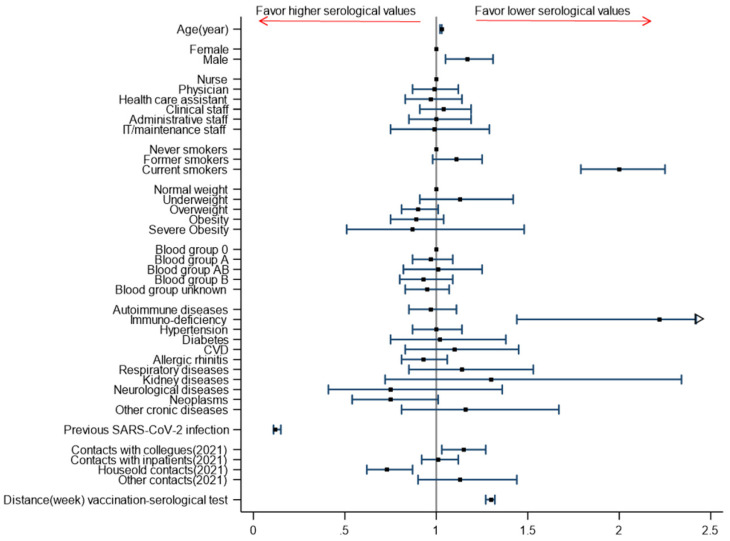
Results of multivariable ordinal logistic regression model (ORs and 95% CI) for predictors of lower serological values (quintiles of distribution) among vaccinated HCW of CSS.

**Figure 2 vaccines-10-00474-f002:**
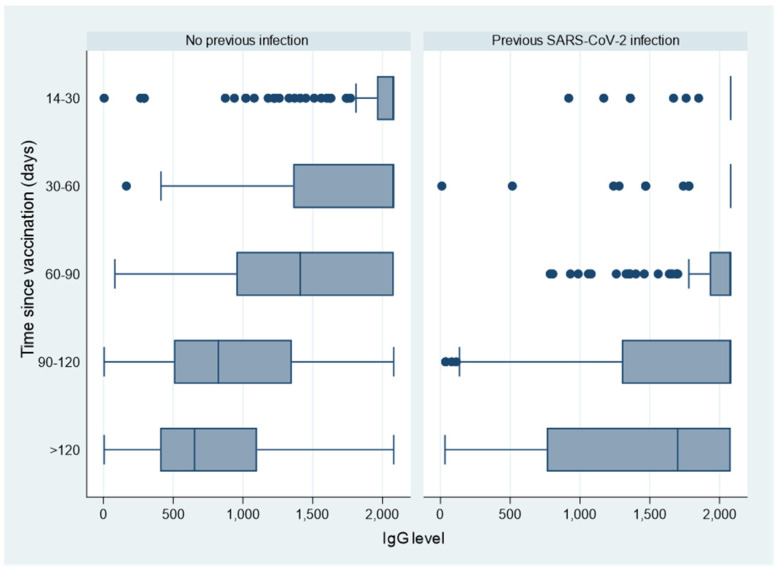
Box-plot of IgG values according to distance (days) between vaccination and serological test on May 2021, by previous SARS-CoV-2 infection.

**Table 1 vaccines-10-00474-t001:** Participants (first column), prevalence of seropositivity (second column), and serological values (third column) among 6687 vaccinated HCW of CSS participating in the serological post-vaccination survey on May 2021, by individual and socio-demographic characteristics.

	Participants and Prevalence of Positivity		Serological Values
	Total—*N* (%)	Serological Positivity—*N* (%)	IgG Value (BAU/mL)—Median (IQR)
OVERALL	6687	6675 (99.8%)	990 (551–1870)
Age (yrs)			
Mean (SD)	47.4 (11.3)		
Median (IQR)	49.5 (38.7–56.5)		
Age-groups (yrs):			
≤29	751 (11.2%)	751 (100.0%)	1360 (821–2060)
30–39	1055 (15.8%)	1055 (100.0%)	1090 (667–1860)
40–49	1643 (24.6%)	1641 (99.9%)	912 (518–1710)
50–59	2417 (36.1%)	2410 (99.7%)	910 (487–1870)
≥60	821 (12.3%)	818 (99.6%)	873 (487–1770)
Gender			
Female	5006 (74.9%)	4997 (99.8%)	1030 (570–1890)
Male	1681 (25.1%)	1678 (99.8%)	874 (511–1740)
Job profile			
Nurse	2315 (34.6%)	2312 (99.9%)	1010 (544–1950)
Physician	1644 (24.6%)	1640 (99.8%)	975 (573–1745)
Health care assistant (HCA)	748 (11.2%)	745 (99.6%)	1020 (511–2080)
Clinical staff (other than physician/nurse/HCA)	1073 (16.0%)	1071 (99.8%)	960 (550–1670)
Administrative staff	692 (10.3%)	692 (100.0%)	981 (553–1810)
IT/maintenance staff	215 (3.2%)	215 (100.0%)	905 (496–1790)
Smoking habit			
Never smokers	4054 (60.6%)	4050 (99.9%)	1100 (624–1990)
Former smokers	1152 (17.2%)	1149 (99.7%)	978 (555–1935)
Current smokers	1481 (22.1%)	1476 (99.7%)	729 (399–1360)
Body Mass Index (BMI)			
Underweight (BMI < 18.5)	296 (4.4%)	295 (99.7%)	973 (493–1550)
Normal weight (BMI18.5–25)	4071 (60.9%)	4066 (99.9%)	996 (555–1820)
Overweight (BMI 25–30)	1626 (24.3%)	1624 (99.9%)	973 (556–1980)
Obesity (BMI 30–40)	641 (9.6%)	639 (99.7%)	970 (528–2070)
Severe obesity (BMI > 40)	53 (0.8%)	51 (96.2%)	1170 (533–2080)
AB0 Blood Group			
0	2375 (35.5%)	2372 (99.9%)	964 (545–1810)
A	1939 (29.0%)	1935 (99.8%)	1050 (571–1930)
AB	328 (4.9%)	326 (99.4%)	923 (500–1850)
B	706 (10.6%)	706 (100.0%)	1000 (553–1840)
Unknown	1339 (20.0%)	1336 (99.8%)	956 (542–1850)

**Table 2 vaccines-10-00474-t002:** Participants (first column), prevalence of seropositivity (second column), and serological values (third column) among 6687 vaccinated HCW of CSS participating in the serological post-vaccination survey on May 2021, by previous SARS-CoV-2 infection, contacts at risk, and distance between vaccination and serological test.

	Participants and Prevalence of Positivity		Serological Values
	Total—*N* (%)	Serological Positivity—*N* (%)	IgG Value (BAU/mL)—Median (IQR)
OVERALL	6687	6675 (99.8%)	990 (551–1870)
Previous SARS-CoV-2 infection *			
No	5465 (81.7%)	5455 (99.8%)	859 (504–1450)
Yes	1222 (18.3%)	1220 (99.8%)	2080 (1340–2080)
Contacts at risk in 2021			
-with collegues			
No	4687 (70.1%)	4678 (99.8%)	1020 (566–1930)
Yes	2000 (29.9%)	1997 (99.9%)	901 (519–1690)
-with patients			
No	3758 (56.2%)	3749 (99.8%)	1000 (543–1880)
Yes	2929 (43.8%)	2926 (99.9%)	971 (559–1860)
-in the household			
No	6186 (92.5%)	6174 (99.8%)	975 (544–1850)
Yes	501 (7.5%)	501 (100.0%)	1100 (661–2080)
-others			
No	6432 (96.2%)	6420 (99.8%)	994 (552–1870)
Yes	255 (3.8%)	255 (100.0%)	942 (545–1570)
Distance (weeks) between vaccination and serological test			
Mean (SD)	14.2 (3.1)		
Median (IQR)	14.9 (13.6–16.0)		
Distance categories:			
2–4 weeks	175 (2.6%)	174 (99.4%)	2080 (2080–2080)
4–8 weeks	260 (3.9%)	259 (99.6%)	2080 (1730–2080)
8–12 weeks	422 (6.3%)	422 (100.0%)	1685 (1060–2080)
12–16 weeks	4214 (63.0%)	4208 (99.9%)	931 (553–1670)
>16 weeks	1513 (22.6%)	1509 (99.7%)	716 (423–1300)
Vaccination date not known	103 (1.5%)	103 (100.0%)	1480 (829–2080)

* From March 2020 to the date of serological survey on May 2021.

**Table 3 vaccines-10-00474-t003:** Participants (first column), prevalence of seropositivity (second column), and serological values (third column) among 6687 vaccinated HCW of CSS participating in the serological post-vaccination survey on May 2021, by self-reported diseases/disorders.

	Participants and Prevalence of Positivity		Serological Values
	Total—*N* (%)	Serological Positivity—*N* (%)	IgG Value (BAU/mL)—Median (IQR)
OVERALL	6687	6675 (99.8%)	990 (551–1870)
Autoimmune diseases			
No	5812 (86.9%)	5806 (99.9%)	987 (552–1860)
Yes	875 (13.1%)	869 (99.3%)	1010 (541–1920)
Immunodeficiency			
No	6609 (98.8%)	6601 (99.9%)	995 (554–1870)
Yes	78 (1.2%)	74 (94.9%)	629 (327–1440)
Hypertension			
No	5660 (84.6%)	5652 (99.9%)	1000 (563–1860)
Yes	1027 (15.4%)	1023 (99.6%)	896 (495–1920)
Diabetes			
No	6530 (97.7%)	6519 (99.8%)	995 (553–1860)
Yes	157 (2.3%)	156 (99.4%)	876 (484–2080)
Cardiovascular diseases			
No	6499 (97.2%)	6487 (99.8%)	995 (555–1860)
Yes	188 (2.8%)	188 (100.0%)	873 (439–1960)
Allergic rhinitis			
No	5801 (86.8%)	5792 (99.8%)	981 (543–1850)
Yes	886 (13.2%)	883 (99.7%)	1040 (598–1950)
Respiratory diseases			
No	6513 (97.4%)	6501 (99.8%)	991 (553–1860)
Yes	174 (2.6%)	174 (100.0%)	935 (481–1880)
Kidney diseases			
No	6647 (99.4%)	6636 (99.8%)	991 (552–1870)
Yes	40 (0.6%)	39 (97.5%)	786 (378–1620)
Neurological diseases			
Yes	6649 (99.4%)	6638 (99.8%)	990 (552–1870)
SI	38 (0.6%)	37 (97.4%)	1150 (483–1530)
Neoplasms			
No	6543 (97.8%)	6532 (99.8%)	990 (552–1860)
Yes	144 (2.2%)	143 (99.3%)	1025 (519–1970)
Other chronic diseases			
No	6582 (98.4%)	6570 (99.8%)	989 (551–1860)
Yes	105 (1.6%)	105 (100.0%)	1110 (585–1930)

## Data Availability

The data presented in this study are available within the article and Appendix A.

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
