# Peer review of "Factors Influencing Level and Persistence of Anti SARS-CoV-2 IgG after BNT162b2 Vaccine: Evidence from a Large Cohort of Healthcare Workers"

_vaccines, 2022, doi:10.3390/vaccines10030474_

Round 1

Reviewer 1 Report

The manuscript is very interesting even if it represents one of the articles already published on the topic.

The reviewer suggests to revise Figure S1 of Appendix to make it more understandable to readers

Author Response

We thank the reviewer and appreciate the considerations. Regarding Figure 1 of Appendix, we would like to keep it in its present detailed form, that is in line with the requirements of the STROBE international guidelines for reporting of observational studies (https://www.equator-network.org/reporting-guidelines/strobe/)

Reviewer 2 Report

Costa et al., analyzed serum anti-SARS-CoV-2 IgG levels from 6687 health care workers and identified factors influencing IgG levels after BNT162B2 vaccination. This study is well designed and conducted. I have only two minor issues.

  1. They have analyzed many independent factors, such as sex, ages, etc. If possible, it is suggested that multivariate analysis can be conducted. The multivariate analysis is able to identify which factors are indeed independent.
  2. Table 1 is difficult to follow with the first two columns are number of individuals and the last is the IgG values. The authors should modify the Table 1 to make it easier to read.

Author Response

1) We thank the reviewer and appreciate the considerations. Actually, we did conduct a multivariable analysis, as reported in the methods section (Statistical analyses, page 4). In order to make it clearer, we revised the sentence of the Results section that introduce the results of the multivariable analysis ( page 7).

2) Tables 1A,B, and C report in fact three types of information, i.e. the number (%) of subjects (first column), the number (%) of positive subjects (second column) and the median (IQR) serological value (third column), for each subgroup category. We have modified the title of the table and the labels to make it clearer. We thought that this was the best way to describe a large amount of essential information in only one table. In order to make the table more readable we could split it into two tables each, but this would double the lentgh of the (already long) Tables 1 A,B, and C . We certainly could do this if the Editor agrees.

Reviewer 3 Report

In their work, Costa et al. evaluated SARS-CoV-2-specific IgG response to BNT162b2 vaccination in a large cohort of healthcare workers and defined factors influencing level and persistence of specific Ab response.

The manuscript is obviously a revised version and it seems to be that the authors made major changes in response to recommendations not known for me.

The current form of the manuscript represents valuable information to define factors being important for optimalization of vaccine strategies.

The only concern from me is the lower serological levels in male gender. The authors demonstrated that increasing age lowers serological levels. Therefore, it should be proofed that the age of male and female participants does not differ and lower serological levels seen in males are not related to higher age in this group.   

Author Response

In their work, Costa et al. evaluated SARS-CoV-2-specific IgG response to BNT162b2 vaccination in a large cohort of healthcare workers and defined factors influencing level and persistence of specific Ab response.

The manuscript is obviously a revised version and it seems to be that the authors made major changes in response to recommendations not known for me.

We thank the reviewer and appreciate the considerations. Actually, the paper was not previously revised by others; we simply erroneously submitted a version with our last revisions tracked. We apologize for this. In the present version of the paper, we accepted all our previous revisions.

The current form of the manuscript represents valuable information to define factors being important for optimalization of vaccine strategies.

The only concern from me is the lower serological levels in male gender. The authors demonstrated that increasing age lowers serological levels. Therefore, it should be proofed that the age of male and female participants does not differ and lower serological levels seen in males are not related to higher age in this group. 

We thank the reviewer for this comment. Actually, the distribution of age in males and females does not differ significantly. In any case, we can exclude a confounding effect by age because the role of sex reported in table S3 and Figure 1 is carefully adjusted by age (included into the multivariable model as a continuous variable) and for all other variables in the table. We have clarified this interpretation in the revised version of the manuscript, underlying that lower serological values among men have been reported in some other studies, and certainly deserves further investigations.